# Marketing to Children in Supermarkets: An Opportunity for Public Policy to Improve Children’s Diets

**DOI:** 10.3390/ijerph17041284

**Published:** 2020-02-17

**Authors:** Jennifer L. Harris, Victoria Webb, Shane J. Sacco, Jennifer L. Pomeranz

**Affiliations:** 1Rudd Center for Food Policy & Obesity, University of Connecticut, Hartford, CT 06103, USA; 2Springfield Psychological, Philadelphia, PA 19102, USA; vwebb@springpsych.com; 3Department of Allied Health Sciences, University of Connecticut, Mansfield, Storrs, CT 06269, USA; shane.sacco@uconn.edu; 4College of Global Public Health, New York University, New York, NY 10003, USA; jlp284@nyu.edu

**Keywords:** food and beverage marketing, in-store marketing, childhood obesity, pester power, sugary drinks, children’s foods, food policy

## Abstract

Public health experts worldwide are calling for a reduction of the marketing of nutrient-poor food and beverages to children. However, industry self-regulation and most government policies do not address in-store marketing, including shelf placement and retail promotions. This paper reports two U.S.-based studies examining the prevalence and potential impact of in-store marketing for nutrient-poor child-targeted products. Study 1 compares the in-store marketing of children’s breakfast cereals with the marketing of other (family/adult) cereals, including shelf space allocation and placement, special displays and promotions, using a national audit of U.S. supermarkets. Child-targeted cereals received more shelf space, middle- and lower-shelf placements, special displays, and promotions compared with other cereals. Study 2 compares the proportion of product sales associated with in-store displays and promotions for child-targeted versus other fruit drinks/juices, using syndicated sales data. A higher proportion of child-targeted drink sales were associated with displays and promotions than sales of other drinks. In both categories, the results were due primarily to major company products. Although in-store marketing of child-targeted products likely appeals to both children and parents, these practices encourage children’s consumption of nutrient-poor food and drinks. If companies will not voluntarily address in-store marketing to children, government policy options are available to limit the marketing of unhealthy foods in the supermarket.

## 1. Introduction

Food marketing is a significant contributor to poor diet and obesity among children, and experts are calling for significant reductions in unhealthy food marketing to children to address this worldwide public health crisis [1,2,3]. In the United States, food and beverage companies spent almost $1 billion on marketing targeted directly at children under 12 in 2009; this was almost entirely on nutrient-poor products, including fast-food restaurants, high-sugar cereals, sugary drinks, snack foods, candy, and desserts [4]. Worldwide, food marketing to children overwhelmingly promotes nutrient-poor foods [3,5].

In most countries, governments have relied upon industry self-regulatory programs to implement improvements in child-targeted food marketing [6,7], and the food industry has responded with promises to market healthier dietary choices to children [8]. Through voluntary industry self-regulatory initiatives, such as the Children’s Food and Beverage Advertising Initiative (CFBAI), in the United States, companies pledge to only advertise healthier dietary choices in “child-directed” media [9]. However, independent evaluations of industry self-regulation demonstrate limited improvements in the amount of food marketing to children, and the majority of advertised products remain high in sugar, fat, sodium, and/or calories [10]. 

Furthermore, the CFBAI and other industry self-regulatory initiatives have been criticized for loopholes that enable companies to continue to market unhealthy foods and beverages to children in ways not covered by CFBAI pledges [11,12]. One significant loophole of the CFBAI and other self-regulatory initiatives is that they do not cover any form of marketing that occurs in retail establishments, including product packaging and in-store marketing. Yet parents identify in-store marketing, such as licensed characters on packaging and unhealthy products placed at children’s eye level, as some of the most common and highly effective types of marketing to children [13,14]. In 2009, U.S. food companies reported spending $72 million on product packaging and marketing at retailers specifically aimed at children, primarily makers of snack foods, cereals, beverages, candy, and frozen desserts [4]. As a result, public health experts have called on food and beverage companies to reduce child-targeted marketing of nutrient-poor products in the supermarket, including through product packaging and in-store marketing practices [11].

However, the majority of research on food marketing to children has focused on traditional forms of media advertising, especially TV advertising [2,3]. In this paper, we begin to address this research gap by summarizing the existing literature on child-targeted marketing in the supermarket. We then present results of two studies comparing U.S. in-store marketing for child-targeted versus other products in the cereal and fruit drinks and juice categories, two types of products commonly marketed to children under 12 in stores [4]. Our primary aims for these studies were to: (1) document the prevalence of in-store marketing for children’s cereals and fruit drinks; (2) examine differences in in-store marketing between children’s products and other similar products; and (3) quantify associations between in-store marketing and product sales. We conclude with a discussion of possible policy interventions to limit unhealthy food marketing in the retail environment.

### 1.1. Child-Targeted Marketing in the Supermarket

Food companies utilize a variety of tactics to market to children in supermarkets, including child-targeted marketing messages on product packages, special displays to attract children’s attention, and even mini events and contests at retailers [4]. Several studies have documented the prevalence of child-targeted packaging in supermarkets [15,16,17,18,19,20]. Common techniques include premiums (e.g., toys, games, contests), brand spokes-characters, licensed characters, and other cross-promotions. Product packages also attract children with bright colors, fun fonts, colorful graphics, and unusual shapes, and through portable, easy-to-use, and durable package designs. These studies also conclude that products high in fat, sugar, and/or sodium most commonly utilize child-targeted techniques on product packaging. 

In addition to product packaging, child-targeted marketing in supermarkets includes strategic shelf placement, special displays (e.g., endcap displays at the end of aisles, free-standing displays or bins in aisles or the front of store), pricing incentives, and samples or tastings [4,21]. Researchers have observed that child-targeted foods are commonly placed on shelves at children’s eye level [15,22]. Dixon and colleagues found that products with promotions or colorful packaging were generally positioned at the lowest point within reach of a child [22]. Supermarkets also dedicate ample shelf space to products that appeal to children. For example, Winson demonstrated that 55%‒80% of shelf space in the cereal aisle was dedicated to pre-sweetened cereals [23]. 

Special displays are also primarily used to market low-nutrient foods. Candy, salty snacks, sugary drinks, and sweetened cereals are often placed in special bins, endcap displays, and checkout lanes [22,23,24,25]. One Australian study found that more than one-third of endcap displays in stores featured snack foods, and stores averaged four free-standing bins that contained chocolate items [25]. A U.S. study showed that supermarkets had significantly more special displays and price reductions for sugary drinks and foods high in saturated fats, added sugar, and sodium compared with nutritious foods [24]. Although supermarket displays for nutrient-poor foods likely attract children’s attention, these studies did not specifically examine children’s products or child-targeted techniques on displays.

“Slotting allowances” represent another form of marketing in supermarkets. Companies report that they often pay a slotting allowance to retailers in order to secure shelf placement for new products or space in checkout lanes [26,27,28]. Concerns about slotting fees focus primarily on their disproportionate use by large manufacturers, which places smaller manufacturers who cannot afford to pay at a competitive disadvantage. The U.S. Federal Trade Commission (FTC) did not include this practice in its analysis of expenditures on marketing to youth [4], but these fees can be substantial [26,27,28].

### 1.2. In-Store Marketing Effectiveness

Numerous studies indicate that children influence their parents’ shopping decisions (often referred to as “pester power”) [29,30,31,32], and mothers report that their children influence food purchases more than any other product [33]. Multiple observational studies and surveys indicate that children’s in-store persuasion attempts are often successful [31,32,34,35], and that children commonly request sweets or snacks [31]. 

Research also has identified in-store marketing tactics that enhance both the likelihood and effectiveness of a child’s “pester power” and its interaction with television advertising to encourage children to request products they have seen advertised [36,37]. For example, families with children exhibited greater variety seeking for children’s cereals and sodas [38], and households with children purchased child-targeted cereals with TV advertising 13 times more frequently than non-advertised cereals [39]. These studies suggest that advertising may persuade children and parents to purchase advertised brands. 

Although few academic studies have examined the effects of in-store marketing overall [40] and no studies have examined direct effects on children, research has shown that special displays, shelf placement, and pricing affect adults’ food purchases. Special displays attract consumers’ attention and are especially effective at increasing unplanned or impulse purchases [41,42]. Displays are more effective than increasing amount of shelf space alone [43]. Cohen and colleagues also found that individuals’ exposure to in-store displays for nutrient-poor foods, including sugary drinks and unhealthy foods, was associated with higher BMI [24]. Another study found that endcap displays increased sales of carbonated drinks by approximately 50% [44].

The amount of shelf space in supermarkets (i.e., “facings” or number of product packages on the shelf “facing” a shopper) also increases purchases of impulse products, but not staples [45]. One eye-tracking study also showed that the number of shelf facings for a brand strongly influenced consumers’ attention toward and evaluation of the brand [46]. However, the number of facings appears to be less important than the shelf placement, with eye-level shelf placement most effective, including for breakfast cereals [47] and snack foods [42]. Therefore, products placed at eye level (i.e., middle shelf for adults) are likely to capture the attention of consumers and thus have the opportunity to influence purchase decisions, especially for unplanned or impulse purchases. On the other hand, studies show that many child-targeted foods are placed on the bottom shelf, at children’s eye level and within their reach [15,22], and children are more likely to request products at their eye level [35]. Although studies with children have not measured the direct effects of shelf placement on product sales, it is likely that additional child requests result in increased sales. 

Surveys of parents often cite product price as another influential component of their purchasing decision [35,48,49]. Intervention studies have lowered the prices of healthier foods in supermarkets, vending machines, and cafeterias to show that discounts increase product sales [50,51,52]. Cohen and colleagues also demonstrated that exposure to in-store price reductions for sugary drinks was associated with higher BMI [24].

Overall, research indicates that in-store marketing, including marketing targeted directly to children, is common and likely impacts family purchases and children’s diets. Numerous studies have documented the extent and content of food packaging that attracts children’s attention in the supermarket. In addition, research with adults demonstrates that eye-level shelf placement, special displays, and pricing incentives motivate purchase, especially for unplanned or impulse purchases. However, few studies have examined these forms of in-store marketing for child-targeted products. 

Given that in-store marketing commonly promotes nutrient-poor foods, likely affects children and their parents, and negatively impacts children’s diets and long-term health, it is important to better understand the extent and impact of marketing for child-targeted products in the supermarket. 

### 1.3. The Present Research

We report results of two research studies that examined in-store marketing for two categories that are commonly marketed to children: breakfast cereals and fruit drinks [4]. Using data from two previous studies, we examined number of facings, shelf placement, special displays, and price promotions for cereals in a national sample of supermarkets, and national sales data for fruit drinks and juices. These studies test the following hypotheses: (1) child-targeted products are more likely to be placed at children’s eye level (i.e., on the bottom shelf or shelves) compared with products marketed to adults only (Study 1); (2) supermarkets disproportionately feature child-targeted products in special displays and price promotions (Studies 1 and 2); and (3) displays and price promotions are associated with higher sales of child-targeted products (Study 2). We also examine whether differences in marketing of child-targeted versus adult products in stores are greater for major food manufacturers compared with smaller manufacturers (Studies 1 and 2). 

## 2. Study 1: Breakfast Cereals 

U.S. cereal companies spent $173 million in marketing to children under 12, more than any other type of packaged food [4]. Expenditures included $5.2 million on marketing in stores; only snack food companies spent more to target children in stores ($6.3 million). In addition, previous studies have documented a clear distinction between children’s cereals (i.e., those marketed directly to children), “family” cereals marketed to parents to serve their children, and adult cereals marketed to adults for their own consumption [53,54]. Furthermore, child-targeted cereals are less nutritious than adult cereals, with 57% more sugar, 50% more sodium, and 52% less fiber [53]. In 2017, the majority of children’s cereals had 9–10 g of sugar per 27–30 g serving [10].

In this study, we compare in-store marketing of children’s breakfast cereals with marketing of family and adult cereals, including shelf space allocation and placement, special displays, and promotions. We also compare results for the two largest cereal manufacturers versus other manufacturers. 

### 2.1. Materials and Methods

Researchers commissioned an audit of cereal marketing in supermarkets using a market research firm specializing in retail research with a nationwide network of experienced field personnel [55]. Descriptive results by cereal and detailed methods have been reported previously [56]. Field representatives visited 400 large supermarkets and Walmart branches in 16 major U.S. cities during May‒June 2009. In week 1, they recorded the number of facings (i.e., number of package fronts on the shelf facing the customer) for each breakfast cereal product (N = 208) and the shelf or shelves on which each cereal was located (top, middle, bottom). If the store had five or six shelves, representatives categorized the top two and bottom two shelves as top and bottom, respectively. 

Representatives then conducted follow-up audits in 87 stores each week for four subsequent weeks. These 87 stores were randomly selected from the original 400 stores to include one store from each supermarket chain in each of the 16 cities. The follow-up audits examined special displays and promotions for cereals. They documented all special displays for cereals and other promotional materials present anywhere within the store. Special displays were classified as: (1) endcaps, displays located at the end of an aisle; (2) in-aisle displays, free-standing manufacturer or case displays located within an aisle; and (3) all other displays located elsewhere in the store, such as the entrance or exit. Promotions were categorized according to the three types used most often by cereal manufacturers in supermarkets [57]: (1) price promotions, special price signage in the aisle that communicated sale prices or special bargains, these could be either store or manufacturer generated; (2) shelf coupon machines, dispensers of manufacturer coupons placed within the aisle; and (3) shelf danglers, signs that hang from a shelf calling attention to a particular item. In addition, all other in-store promotions not fitting into these categories, such as floor graphics or shopping-cart advertisements, were recorded. 

Using the week 1 data, researchers calculated the following dependent variables for analysis: percent of stores stocking each cereal; the average number of facings per store stocking it; and the percent of stores stocking the cereal on the bottom, middle, and top shelves. Using the four-week data, researchers calculated additional dependent variables for the average number of stores featuring each type of display and promotion per cereal over the four-week period (total number of displays/promotions during the four weeks for each cereal divided by the number of stores stocking the cereal).

All cereal products were categorized according to target audience (child, family or adult) and company type (major or other). Target audience was identified using data collected for a comprehensive study of cereal marketing to children [56]. Cereals designated as child-targeted had some type of marketing aimed directly at children, such as child-targeted TV advertising or websites, or featured a children’s licensed character. Family products included all other brands with any marketing suggesting that it was appropriate to serve to children, but that did not utilize other forms of marketing appealing directly to children. All other products were designated as adult products. None of the marketing materials for adult cereals indicated that children should or would want to consume the product. 

Products from the two largest cereal manufacturers (General Mills and Kellogg) were classified as major company products. These manufacturers stocked their products in 100% of supermarkets and represented more than one-half of total cereal shelf facings [56]. In addition, the two companies spent over $300 million to advertise their products in 2008, which represented 93% of all advertising spending for the cereal category. All other companies were classified as other company. Although Kellogg and General Mills own Kashi and Cascadian Farms, they were classified as separate companies as their association with the parent company did not appear in any of their marketing and was not readily apparent to the consumer. 

We conducted a series of two (company type) by three (target audience) multivariate analyses of variance (MANOVAs) to evaluate differences in: (1) number of facings per cereal; percent of stores stocking the cereal; and percent stocking it on bottom, middle and top shelves; (2) total number of special displays including endcap, in-aisle, and all other; and (3) total number of promotions, including price promotions, coupons, danglers, and all other. To examine significant interactions, we conducted one-way MANOVAs for target audience, looking at major and other companies separately. Significant differences between individual target audiences were assessed using the Bonferroni procedure. Means and 95% confidence intervals (CIs) are reported.

### 2.2. Results

Of the 208 cereals examined, 23% qualified as child-targeted, 35% targeted families, and 42% targeted adults only. Overall, the two major companies produced 54% of cereals in the analysis, but this proportion varied by target audience, χ^2^ (2, *N* = 208) = 12.2, *p* = 0.002. Major companies offered 65% of child-targeted cereals and 64% of family cereals, but just 40% of adult-targeted cereals. 

#### 2.2.1. Percent Stocking and Shelf Facings 

Stores were significantly more likely to stock cereals from major companies (*M*_m_ = 54% (48, 61) of stores stocking) than cereals from other companies (*M*_o_ = 40% (32, 48)), *F*(1202) = 7.9, *p* = 0.006. A higher percentage of stores stocked adult cereals (*M*_a_ = 54% (47, 62)) than family cereals *M*_f_ = 43% (35, 51)) and child-targeted cereals (*M*_c_ = 44%, (34, 54)), but the difference was not significant, *F*(2202) = 2.6, *p* = 0.08. In addition, the interaction between company type and target audience for percent stocking was not significant (*p* = 0.30).

Stores also devoted more facings to major-company cereals (*M*_m_ = 2.6 (2.4, 2.9) facings per cereal) than to other-company cereals (*M*_o_ = 2.1 (1.8, 2.4)), *F*(1, 202) = 7.1, *p* = 0.008. The number of facings per cereal also differed by target audience, *F*(2, 202) = 3.8, *p* = 0.03. Stores averaged more facings for child-targeted cereals (*M*_c_ = 2.7, (2.3, 3.1)) than family cereals (*M*_f_ = 2.0 (1.7, 2.3), *p* = 0.02), but not adult cereals (*M*_a_ = 2.4 (2.1, 2.6), *p* = 0.47). 

The interaction between company type and target audience was marginally significant for number of facings, *F*(2, 202) = 2.9, *p* = 0.06. For cereals from major companies, number of facings differed by target audience, *F*(2, 109) = 4.5, *p* = 0.01, with stores devoting more facings to child-targeted cereals (*M*_c_ = 3.2 [2.7, 3.8]) compared with family (*M*_f_ = 2.3 (1.9, 2.7), *p* = 0.02) and adult cereals (*M*_a_ = 2.3 (1.9, 2.8), *p* = 0.04) from major companies. The number of facings also differed by target audience for cereals from other companies, *F*(2, 93) = 3.3, *p* = 0.04, but stores devoted more facings to adult cereals (*M*_a_ = 2.4 (2.1, 2.7)) than to family cereals (*M*_f_ = 1.8 (1.4, 2.2), *p* = 0.04). However, facings for other-company child-targeted cereals (*M*_c_ = 2.2 (1.7, 2.7)) did not differ from their adult or family cereals (all *p* > 0.61).

#### 2.2.2. Shelf Placement

Stores were significantly more likely to place cereals from major companies (*M*_m_ = 53% (49, 57)) on the prime middle shelf, compared with other companies’ cereals (*M*_o_ = 43% (39, 48)), *F*(1202) = 10.7, *p* = 0.001. They also placed fewer major-company cereals on the bottom shelf (*M*_m_ = 22% (18, 25)) versus other-company cereals (*M*_o_ = 32% (27, 36)), *F*(1, 202) = 12.0, *p* = 0.001. Top-shelf stocking did not differ by company type (*p* = 0.78). 

The percent of stores stocking cereals on the bottom, middle, and top shelves also differed by target audience, *F*(2, 202) = 21.7, 22.5, and 75.9, respectively, all *p* < 0.001. Stores were significantly more likely to place child-targeted and family cereals on the prime middle shelves (*M*_c_ = 52% (46, 57); *M*_f_ = 57% (52, 61)) compared with adult cereals (*M*_a_ = 36% (32, 40), *p* < 0.001); but middle-shelf stocking did not differ between child and family cereals (*p* = 0.52). Stores were also significantly more likely to stock child-targeted cereals on bottom shelves (*M*_c_ = 38%, (32, 44)) compared to family and adult cereals (*M*_f_ = 27% (22, 31), *p* = 0.008; *M*_a_ = 15% (11, 19), *p* < 0.001); family cereals were also more likely to be stocked on bottom shelves than adult cereals (*p* = 0.001). In contrast, significantly more stores stocked adult cereals (53% (48, 52)) on top shelves, compared with child and family cereals (*M*_c_ = 14% [8,20]; *M*_f_ = 19% (14, 23); *p* < 0.001); but the percent stocking child and family cereals on top shelves did not differ (*p* = 0.68). 

Figure 1 illustrates the interaction between type of company and target audience for percent of supermarkets stocking cereals on top, middle and bottom shelves. Means and 95% CIs are reported. Major companies include General Mills and Kellogg. Percent of supermarkets that stocked a cereal can exceed 100% because some stores stocked cereals on more than one shelf in a store.

Interactions between company type and target audience were also significant for the middle and top shelves (*F*(2, 202) = 29.5 and 19.5, respectively; *p* < 0.001), but not for the bottom shelf (*p* = 0.86). For major-company cereals only, supermarkets stocked the majority of child and family cereals on the middle shelves, whereas middle-shelf stocking did not differ by target audience for cereals from other companies (see Figure 1). For both types of companies, stores were more likely to stock child-targeted cereals on bottom shelves and adult cereals on top shelves, compared to cereals targeting other audiences. 

#### 2.2.3. Special Displays and Promotions

On average, 75% of major-company cereals and 59% of other-company cereals were featured in any type of special display during the four weeks analyzed. Endcaps represented approximately three-quarters of displays for cereals from both types of companies. Major company cereals had more total displays (*M*_m_ = 0.27, (0.20, 0.34) vs. *M*_o_ = 0.13, (0.05, 0.22)), *F*(1, 202) = 5.7, *p* = 0.02; endcaps (*M*_m_ = 0.22, (0.15, 0.29) vs. *M*_o_ = 10, (0.02, 0.18)), *F*(1, 202) = 5.1, *p* = 0.03; and other displays (*M*_m_ = 0.03, (0.03, 0.04) vs. *M*_o_ = 0.02, (0.01, 0.03)), *F*(1, 202) = 7.7, *p* = 0.006, compared to other companies, but in-aisle displays did not differ by company type (*p* = 0.41).

Figure 2 presents the average number of displays per cereal over four weeks, including endcaps, in-aisle and other displays, by company type and cereal target audience. Means and 95% CIs are reported for number of total displays and endcaps. Major companies include General Mills and Kellogg.

The number of total displays, endcaps and in-aisle displays differed by target audience, *F*(2, 202) = 5.3, 4.2 and 6.9, *p* = 0.006, 0.02 and 0.001, respectively, with child cereals featured in the most displays of all types. The number of endcaps for child cereals (*M*_c_ = 0.28 (0.17, 0.38)) was almost three times higher than for family (*M*_f_ = 0.10 (0.01, 0.19), *p* = 0.03) and adult cereals (*M*_a_ = 0.10 (0.03, 0.18), *p* = 0.03); and in-aisle displays for child cereals (*M*_c_ = 0.03, (0.02, 0.04)) were significantly higher than for family (*M*_f_ = 0.01, (0.00, 0.02), *p* = 0.006) and adult cereals (*M*_a_ = 0.01, (0.00, 0.01), *p* < 0.001). However, the number of endcaps, in-aisle displays, and other displays did not differ for family and adult cereals (all *p* = 0.99). The number of other displays did not differ by target audience (*p* = 0.23). Interactions between target audience and company type for all display types were non-significant (all *p* > 0.16) (see Figure 2).

Cereals from major companies averaged marginally more total promotions per cereal over the four weeks (*M*_m_ = 1.5 [(1.3, 1.7)) than other-company cereals (*M*_o_ = 1.2 (0.9, 1.4)), *F*(1, 202) = 3.7, *p* = 0.06. For all companies, price promotions made up approximately 90% of total promotions. Due to low incidence of coupon machines, shelf danglers and other promotions, we combined these types as other promotions in the analysis.

Figure 3 presents the average number of price promotions and all other promotions per cereal over four weeks by company type and cereal target audience. Means and 95% CIs are reported. Major companies include General Mills and Kellogg. 

Children’s cereals averaged 1.6 (1.3, 2.0) total promotions over the four weeks, while family cereals averaged 1.2 (0.9, 1.5), and adult cereals also averaged 1.2 (1.0, 1.5). However, the number of promotions per cereal, including total promotions, price promotions and all other promotions, did not differ significantly by target audience (all *p* > 0.10). Similarly, interactions between target audience and company type for all promotion types were not significant (*p* > 0.19) (see Figure 3).

### 2.3. Discussion

Stores were more likely to stock cereals from major companies than from other companies, and they allocated more facings to major-company cereals and stocked a higher proportion on prime middle shelves. Child-targeted cereals also received prime shelf placement. Although stores were not more likely to stock child-targeted cereals, they devoted significantly more shelf space (i.e., number of facings) to child cereals than to family and adult cereals. As hypothesized, stores were more likely to stock child cereals on the middle shelf, compared to adult cereals (but not family cereals). Stores were also more likely to stock children’s cereals on the bottom shelves, at children’s eye level, whereas adult cereals were more likely to be placed on top shelves, the position that receives the least attention. However, some of these results differed by company type. Although child-targeted cereals from major companies received better shelf placement than major-company adult cereals, adult cereals from other companies received more facings and were more likely to be placed on the prime middle shelf compared to child cereals from other companies.

Endcaps and price promotions represented the majority of special displays and promotions featuring cereals in supermarkets. As hypothesized, child-targeted cereals also received significantly more marketing support in stores through special displays, compared with adult and family cereals. On average, child-targeted cereals were featured in some type of display in almost one-half of stores during the four weeks examined, more than double the total displays for family and adult cereals, and almost three times the number of endcaps. Child-targeted cereals also averaged 1.6 price promotions over the four weeks, but this number was not significantly higher than price promotions for family and adult cereals.

As expected, major companies’ cereals also received significantly more in-store marketing support, including displays and price promotions, than other companies’ cereals. For both types of companies, child-targeted cereals were featured in more displays, but not more price promotions, compared to the same companies’ adult and family cereals. 

## 3. Study 2: Fruit Drinks and Juices

Study 2 examines incremental product sales attributed to in-store marketing for fruit drinks (fruit-flavored or juice drinks that contain added sugar) and juices (100% juice and juice/water blends without added sweeteners), including child-targeted and other products. In 2009, beverage companies reported spending over $43 million on marketing of juices and other noncarbonated beverages to children under 12, including approximately $3 million (5% of expenditures) on in-store marketing [4]. As with cereals, sugar-sweetened fruit drinks are commonly advertised to both children and adults, while children’s 100% fruit juices and juice/water blends (without added sweeteners) are more often advertised to adults only [58]. Although the American Academy of Pediatrics (AAP) and other key nutrition and health organizations recommend that children never consume drinks with added sugar [59], sugary drinks contribute almost half of all added sugar consumed by children [60]. Furthermore, fruit drinks are the most common type, consumed by approximately one-quarter of two- to 11-year-olds on a given day [60]. In 2018, sugar-sweetened fruit drinks represented 62% of children’s drinks sold in the United States [58]. The AAP also recommends limiting children’s consumption of juice, but concludes that one serving per day can be provided as part of a healthy diet [11].

In this study, we use syndicated sales data to compare the proportion of product sales associated with in-store displays and promotions for child-targeted versus other (i.e., products not marketed directly to children) fruit drinks and juices. We also predict a greater difference in sales due to in-store marketing for child-targeted versus other products from major companies relative to smaller ones. Lastly, we predict that these differences will be greater for products with added sugar (i.e., fruit/juice drinks) compared to those without added sweeteners (i.e., 100% juices, juice/water blends).

### 3.1. Materials and Methods

To track beverage sales, we acquired data from IRi Worldwide, a company that provides marketing and shopper data primarily to retailers and product manufacturers [61]. Specifically, we utilized the company’s scanner-based tracking service, which provides weekly sales data for a sample of food stores (including grocery stores with over $2 million in total sales) for all products scanned in those stores in a given week. IRi also identifies sales that occurred under promotional conditions, including: (1) displays (endcap, lobby, and shipper free-standing [i.e., in-aisle] displays); and (2) temporary price reductions. To identify sales due to displays, IRi representatives conduct weekly store audits to identify products featured in different types of in-store displays that week. In its dataset, IRi reports sales of products that occurred in stores with each type of display in a given week. To identify sales due to price promotions, IRi examines weekly pricing data for each product and store and identifies products with a temporary price reduction, defined as a reduction of 5% or more from the item’s base price for that store. It also reports sales of products that occurred in stores with a temporary price reduction in that week. We used these variables to identify incremental sales that can be attributed to displays and price reductions. These measures were developed and primarily used by manufacturers and retailers to monitor their competitors’ in-store marketing activities.

We licensed sales data for bottled juices (including fruit drinks) and aseptic juices and drinks (i.e., products in box or pouch containers) for supermarkets in eight U.S. cities of various sizes and geographic regions. We received data for all universal product codes (UPCs) with more than $500,000 in sales nationwide for the calendar year 2010. For each UPC, we received total sales in dollars and units (i.e., packages), as well as sales due to displays and price reductions. To determine the percentage of sales due to displays, we divided sales that occurred in stores with displays and incremental sales due to price promotions by the product’s total sales.

Children’s fruit drinks and juices typically come in 4- to 8-ounce single-serving containers such as aseptic juice boxes and pouches, whereas other products (not specifically for children) often come in larger sized multi-serving bottles, as well as single-serve containers [58]. Therefore, we examined sales for single-serving products only. Products sold in a 20-ounce container or smaller and packages of multiple single-serving containers were categorized as single-serving items. We excluded private-label brands.

Products also were categorized according to target audience. Items were coded as child-targeted if they belonged to a brand marketed exclusively to children (i.e., Capri Sun, Kool Aid, Hawaiian Punch, Hi C, and Sunny D) as identified in a prior study of child-targeted beverages [62]. All other products (including products aimed at parents, but not marketed directly to children) were coded as other. Additionally, products were categorized according to the type of manufacturer. Companies that advertised any of their brands on national television in 2010 [62] were categorized as major companies, whereas those that did not engage in national television advertising were labeled as other companies. Finally, we categorized items according to whether they contained added sugar or no added sweeteners (including added sugars and non-nutritive sweeteners).

We conducted two (target audience: child vs. other) by two (company type: major vs. other) by two (added sugar: added sugar vs. no added sweeteners) three-way MANOVAs to evaluate differences in incremental sales for displays and price promotions. Non-parametric bootstrapping was performed [63]. We used unit sales as the outcome variables to capture the relationship between promotions and the number of packages sold, as price promotions could mask associations with dollar sales. We report means with 95% CIs.

### 3.2. Results

Of the 2321 single-serving fruit drink and juice UPCs in the dataset, one-third (32%) qualified as child-targeted, and major companies manufactured two-thirds (64%) of products examined (see Table 1). More than half of the products (54%) contained added sugar and 38% contained no added sweeteners (including 100% juice, juice blends, and juice diluted with water). Another 8% were diet drinks that contained non-nutritive sweeteners with no added sugar. As none of these products (*n* = 185) were targeted to children, they were removed from the analyses. Significantly more child-targeted products contained added sugar (69%) compared to other products (52%), *χ*^2^ = 57.57 (1, *N* = 2136), *p* < 0.001.

We found the predicted main effect of target audience, such that a higher percentage of unit sales for child-targeted products was due to displays (6.5% (5.7, 7.2)) than for other products (5.6% (5.1, 6.2)), *F*(1, 2128) = 3.2, *p* = 0.006. Proportion of sales due to price reductions also was higher for child-targeted products (13.9%, (12.8, 15.1)) than for other products (12.1% (11.3, 12.9)), *F*(1, 2128) = 6.5, *p* = 0.007. In addition, there was a significant main effect of company type on sales due to displays, with a higher percentage of other-company sales due to displays (7.3% [6.6, 8.0]) than for major companies (4.8% (4.3, 5.4)), *F*(1, 2128) = 26.9, *p* = 0.007. However, proportion of sales from price reductions did not differ by company type, *p* = 0.78. Additionally, we observed a significant main effect of added sugar on sales due to displays and price reductions. Products with added sugar had a higher percentage of sales due to displays (9.6% (9.1, 10.2)) compared to products without added sweeteners (2.5% (1.8, 3.3)), *F*(1, 2128) = 227.0, *p* < 0.001, as well as price reductions (17.1% (16.3, 17.9) vs. 8.9% (7.8, 10.0)), *F*(1, 2128) = 133.2, *p* < 0.001.

In addition to these main effects, we observed significant two-way interactions between target audience and company type for percentage of sales due to displays, *F*(1, 2128) = 20.5, *p* = 0.01, and price reductions, *F*(1, 2128) = 19.9, *p* = 0.006; between target audience and added sugar for displays, *F*(1, 2128) = 7.3, *p* = 0.01, but not price reductions (*p* = 0.51); and between company type and added sugar for displays, *F*(1, 2128) = 4.2, *p* = 0.03, but not price reductions (*p* = 0.18). Furthermore, these effects were qualified by a significant three-way interaction between target audience, company type and added sugar for both displays, *F*(1, 2128) = 36.9, *p* = 0.003, and price reductions, *F*(1, 2128) = 39.1, *p* = 0.003. In order to understand these interactions, we conducted two-way MANOVAs for target audience and added sugar, with separate models for major and other companies (see Figure 4).

Figure 4 illustrates the three-way interaction between fruit drink and 100% juice products by company type (major vs. other), target audience (child-targeted vs. other) and sweeteners (added sugar vs. no add sweeteners). Major companies include companies that advertised their brands on national television.

For major companies, child-targeted products had a significantly higher percentage of sales due to displays (6.3% (5.5, 7.2)) versus other products (3.4% (2.9, 3.9)), *F*(1, 1360) = 36.7, *p* < 0.001, and a higher percentage due to price reductions (15.5% (14.1, 17.0) vs. 10.6% (9.7, 11.4)), *F*(1, 1360) = 34.1, *p* < 0.001. Similarly, products with added sugar had a significantly higher percentage of sales due to displays (7.9% (7.3, 8.5)) than products without added sweeteners (1.8% (1.0, 2.6)), *F*(1, 1360) = 156.2, *p* < 0.001, and a higher percentage due to price reductions (17.1% (16.1, 18.0) vs. 9.0% (7.7, 10.4), *F*(1, 1360) = 88.4, *p* < 0.001.

Additionally, we observed a significant interaction between target audience and added sugar for percentage of sales due to displays, *F*(1, 1360) = 70.9, *p* = 0.001, and price reductions, *F*(1, 1360) = 25.1, *p* = 0.01. The higher proportion of sales of added sugar products due to in-store marketing was due primarily to differences for child-targeted products. Child-targeted products with added sugar had more than double the proportion of sales due to displays and price reductions compared to other added-sugar products. Differences between targeted audiences for products without added sweeteners were significantly lower.

For the other companies, the main effects of target audience on sales due to displays and price reductions were not significant (*p* > 0.59). However, we again observed main effects for added sugar, such that percentage of sales due to displays was almost four times higher for added-sugar products (11.3% (10.2, 12.4)) than for products without added sweeteners (3.3% (1.8, 4.7)), *F*(1, 768) = 77.8, *p* < 0.001, and sales due to price reductions was more than twice as high (17.2%(15.7, 18.6) vs. 8.8% (6.9, 10.7)), *F*(1, 768) = 49.5, *p* < 0.001.

We also observed a significant interaction between target audience and added sugar for percentage of sales due to price reductions for other companies, *F*(1, 768) = 15.0, *p* = 0.002, but not for displays (*p* = 0.10). Also, in contrast to the major companies, child-targeted products with added sugar from other companies had a lower percentage of sales due to price reductions compared to other (not child-targeted) products. As found for major companies, the difference between target audiences in sales due to prices reductions for products without added sweeteners was lower.

### 3.3. Discussion

Child-targeted single-serving fruit drinks and juices had a higher percentage of unit sales due to displays and price reductions overall than did similar drinks not targeted to children. A higher percentage of sales of products with added sugar were also due to displays and price reductions compared to products without added sweeteners. These findings indicate that in-store marketing may have a greater impact on sales of drink brands with marketing targeted to children compared to drinks marketed to adults only, as well as on sales of sugary drinks (i.e., fruit drinks in this analysis) versus healthier 100% juices and other juice drinks without added sweeteners.

In contrast to predictions, sales due to displays was lower for major companies (i.e., companies that did not advertise nationally), while sales due to price reductions did not vary by company type. Although not expected, this finding indicates that additional visibility in the supermarket (i.e., through displays) may be more important for products that consumers have not been introduced to through advertising. However, examining the interactions between factors identified some additional differences by company type. For example, sales of child-targeted products from major companies were more likely to be due to displays and price reductions than sales of their other products, whereas differences between child-targeted and other products from other companies were not significant.

These relationships also differed by product type. Although a higher proportion of sales of products with added sugar were due to displays and price reductions for both major and other companies, the interaction with target audience differed by company type. For major companies, sales of sugar-sweetened child-targeted products were most likely to be due to displays and price reductions, compared to other drinks from these companies. In contrast, sales of other sugar-sweetened products (i.e., not child-targeted) from other companies were most likely to be due to displays and price reductions. Thus, in-store displays and price promotions appeared to have the greatest impact on the sales of sugar-sweetened products, including child-targeted fruit drinks, from major companies (e.g., Capri Sun Original [juice drink], Kool-Aid, and Sunny D).

Further research is needed to determine whether major companies promoted sugar-sweetened child-targeted fruit drinks more often in stores (i.e., through displays and price reductions) than they promoted other products or whether these products required additional retail support to drive sales. Regardless, displays and price reductions of sugar-sweetened child-targeted drinks from major companies were associated with 22% and 12% of sales for these products, respectively, compared with just 9% and 1% of sales for major companies’ child-targeted drinks without added sweeteners (primarily 100% juice products), and 12% and 4% of their sugar-sweetened drinks that were not marketed directly to children.

## 4. General Discussion

Together, these studies demonstrate extensive marketing of nutrient-poor child-targeted products in the supermarket and provide evidence that a substantial proportion of sales can be attributed to in-store marketing through displays and price promotions. In Study 1, the two major cereal manufacturers utilized disproportionately more in-store displays and promotions for children’s cereals than for cereals marketed only for adult consumption. Furthermore, major-company children’s cereals received prime middle-shelf placement on supermarket shelves, while children’s cereals from all companies were more likely to be placed on bottom shelves at children’s eye level. Study 2 demonstrated that a higher proportion of sales of sugar-sweetened child-targeted fruit drinks from major companies were due to in-store marketing (special displays and price reductions) compared to sales of other sugar-sweetened drinks not marketed directly to children, as well as child-targeted juices without added sweeteners. In both studies, in-store marketing of nutrient-poor children’s products was greater for major companies than for other companies. These major cereal companies (General Mills and Kellogg) and fruit drink/juice companies that support their products with advertising also likely have more resources to influence promotions and placements in supermarkets.

This evidence of extensive in-store marketing for child-targeted products is problematic due to the poor nutritional quality of these products. Both child-targeted cereals and fruit drinks have worse overall nutritional profiles than products aimed at adults [53,58]. This finding is not unique to the cereal and fruit drinks categories. A large body of literature documents the consistently poor and disproportionately worse nutritional quality of food and beverages marketed to children [3]. Furthermore, despite implementation of food industry self-regulatory initiatives worldwide and repeated promises by industry to market healthier foods to children, independent evaluations of industry self-regulation demonstrate minimal improvements [3]. This lack of industry-led progress provided the basis for the WHO to endorse a set of recommendations for countries to take actions to reduce the negative impact food marketing to children [1]. The FTC report also highlighted the poor nutritional quality of products in other food and beverage categories marketed directly to children in stores in the United States, including snack foods, candy and frozen desserts, and carbonated beverages [4].

One limitation of this research is that data were collected in 2009 and 2010 for previous studies that reported top-line findings [56,62]. Due to the considerable expense of conducting in-store audits and purchasing syndicated sales data, we were not able to purchase more recent data for these studies. However, these findings remain relevant as recent comprehensive analyses of the nutrition quality and child-directed marketing of cereals [10,53] and children’s drinks [58] demonstrate few changes in the nutrition quality of child-targeted products or in the ways they are marketed to children and their parents.

## 5. Policy Implications

Public health experts have emphasized the positive role that supermarkets can play in helping to reverse the childhood obesity epidemic due to the variety of healthier foods that they offer [64]. However, these findings also support the need for public health policy to address unhealthy marketing directed to children in supermarkets and other types of retailers. Potential policy options include both industry self-regulation and government regulation.

### 5.1. Industry Self-Regulation

Food and beverage manufacturers and trade associations have promised to advertise healthier products to children [9,65]. The Grocery Manufacturers Association and some individual retailers have also implemented programs to encourage healthier choices in the supermarket, such as introducing the Facts Up Front nutrition label to help consumers make informed choices and funding nutrition and health-related community programs [66]. Individual retailers, including Walmart and Hannaford, have introduced nutrition rating systems to help all consumers easily identify healthier choices [67,68].

Public health advocates have also partnered with retailers and manufacturers to promote healthy foods to children in supermarkets and other food retailers. For example, the Produce Marketing Association partnered with Sesame Workshop to make its characters available for the promotion of fresh fruits and vegetables [69]. The Partnership for a Healthier American has also partnered with industry and public health organizations to support “FNV,” a campaign that incorporates youth-targeted celebrity endorsements to encourage fruit and vegetable consumption at retailers [70]. The Food Trust, in coordination with local communities, has developed a number of initiatives to encourage availability and marketing of healthier choices in corner stores in neighborhoods without supermarket access [71]. Public health campaigns also call for removing unhealthy food and drinks from checkout aisles at supermarkets and other retailers [72,73].

However, noticeably absent from industry self-regulatory initiatives and public health partnerships are actions to reduce in-store marketing techniques commonly used to promote child-targeted products, including product packaging, special displays, price reductions, and other types of promotions, or to limit in-store access and availability of nutrient-poor foods that appeal to children beyond the checkout aisle.

Food industry representatives argue that self-regulatory initiatives to improve children’s diets and reduce childhood obesity should not address in-store marketing because parents make the final decision about their supermarket purchases [11]. Indeed, our analyses do not examine parents’ decision-making process when choosing to purchase child-targeted products. However, it is likely that in-store marketing of children’s products also influences parents—for example, by attracting their attention and reminding them that their child has asked for the product, or by giving them permission to buy the product when it is on sale. Furthermore, nutrition-related claims on product packaging and other parent-directed marketing techniques often lead parents to believe that high-sugar products, including children’s cereals [74] and fruit drinks [75], are healthy options for their children. Regardless of the mechanism, food manufacturers would not invest in this level of in-store marketing for nutrient-poor children’s products unless it increases sales.

### 5.2. Government Regulation

Given that reducing in-store marketing would likely affect a significant revenue source for both food manufacturers and retailers, government action may be required to effectively reduce the amount of in-store marketing of nutrient-poor child-targeted products. However, in the United States, the government faces substantial limitations to how it can regulate marketing communications, due to protection of marketing under the First Amendment as commercial speech. In order for a retail regulation to not violate the First Amendment, it must be “unrelated to expression” [76], meaning that the advertising content itself (e.g., child-targeted features on a special display) cannot be restricted. However, the government could enact regulations aimed at other characteristics of food products, such as nutrient profiles.

In addition, U.S. states have the authority to enact laws to protect and promote health, safety, and the general welfare of their population that could be used to regulate other practices in the retail environment [77,78]. Under this “police power,” state (and local governments if authorized) can regulate the retail environment and sale of products through direct controls and licensing requirements. States can also influence purchase decisions through their powers to tax and spend and to convey factual information. Therefore, U.S. states have a number of legally viable policy options to regulate in-store marketing (see Table 2). These policy strategies could also be applied in other countries with similar retail environments.

First, states may enact non-speech related restrictions that aim to protect youth, such as age limits on children’s ability to purchase harmful products. For example, one county in New York banned the sale of energy drinks to youth under eighteen years old in county parks [79]. This could also be accomplished as a state-wide age restriction.

The government may address the location of unhealthy food products in stores to positively shape the retail environment for all shoppers, as long as the rationale behind the prohibition is unrelated to communication on the packaging or display (i.e., commercial speech). Therefore, states could regulate placement in the supermarket, such as requiring certain products to be located behind the counter or in a separate aisle. One method to do this is to target food products according to their nutritional profile (e.g., high-sugar products) and apply the nutrition criteria regulation to entire product categories. If the government properly tailored the regulation to address the nutrition quality of products, this would not be considered regulation of speech. These same considerations could apply to restrictions on the placement of products in endcaps, at store entrances, freestanding displays in aisles, and in checkout lanes.

State governments could accomplish the same objectives through conditional licensing, whereby a retailer must obtain a license to operate a supermarket on the condition that it abides by directives specified in the licensing requirements [80]. For example, tobacco retailers often must agree to certain conditions such as posting signs regarding the legal age of purchase, not selling single cigarettes, and not selling tobacco in vending machines or certain locations [81]. One benefit to this strategy is that the license fee can cover the cost of inspection, which is a built-in method of enforcement [80]. Moreover, the threat of losing one’s license is often a stronger deterrent than the threat of a fine alone, so conditional licensing can encourage compliance.

Lastly, government regulation could address incentives and promotions that reduce the price of unhealthy products, such as enacting excise taxes. An excise tax is levied upon businesses engaged in the manufacture, distribution, or sale of commodities. The goal of the tax would be to deter purchase and consumption and additionally raise revenue for the government. The revenue from an excise tax can be earmarked, or dedicated to a certain program such as subsidizing healthy foods. As of February 2017, two U.S. states, one U.S. tribal government, and seven U.S. municipalities taxed junk food or sugary beverage using various taxing mechanisms [82].

The government could also enact minimum price mandates and prohibitions on coupons and discounting. Federal courts have upheld municipal laws that prohibited retailers from redeeming coupons and multipack discounts or prohibited the sale of tobacco products below the listed price [83,84]. This strategy is viable for unhealthy food products and could be enacted independently or in conjunction with a tax to ensure the cost of the tax is passed on to consumers through price increases [80]. Such a mandate or prohibition would counteract food manufacturer pricing strategies that reduce the price of unhealthy products and function to nullify the impact of any tax.

Outside the United States, sugary drink taxes are also increasingly common, and early evaluations show that they effectively reduce the consumption of taxed products [85]. However, despite WHO calls for regulation of unhealthy food marketing targeted to children [1], few countries regulate child-targeted marketing in any form, and current regulations primarily address TV advertising and marketing in schools [7]. One country, Chile, implemented a law that prohibits child-targeted messages (including licensed characters and brand spokes-characters) on product packaging for foods high in fat, sugar, salt, or calories in 2016. To our knowledge, no other country regulates in-store marketing of child-targeted food and drinks in any form.

Public health experts call on food manufacturers and retailers to utilize the power of the supermarket environment to encourage more purchases of fruits and vegetables [86], as well as healthier packaged food options [64]. However, no country currently regulates the common in-store marketing techniques for child-targeted products examined in this paper, including shelf placement, special displays, and price promotions [7].

## 6. Conclusions

Together, these two studies demonstrate that cereal and fruit drink/juice manufacturers engage in extensive in-store marketing-including strategic shelf placement, special displays, and price promotions to promote child-targeted products in supermarkets. Furthermore, major manufacturers utilize these practices significantly more often for child-targeted compared with other products. The analysis of fruit drink/juice products also demonstrates that in-store marketing is associated with higher incremental sales for sugar-sweetened child-targeted products from major companies compared with their other products, including products without added sweeteners and those not targeted to children. Unfortunately, child-targeted products in both categories are high in added sugar and contribute to children’s unhealthy levels of sugar consumption [60]. These findings support public health concerns that marketing in supermarkets contributes to the epidemic of childhood obesity and that government regulation should be enacted to improve the supermarket environment to promote healthy diets for children.

## Figures and Tables

**Figure 1 ijerph-17-01284-f001:**
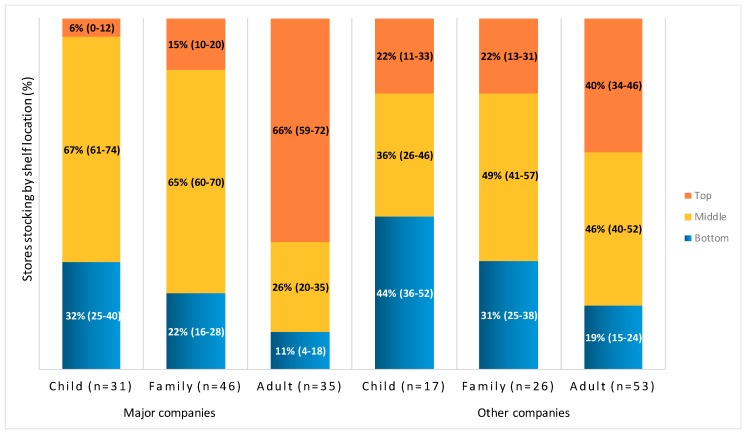
Percent of supermarkets stocking cereals by shelf location.

**Figure 2 ijerph-17-01284-f002:**
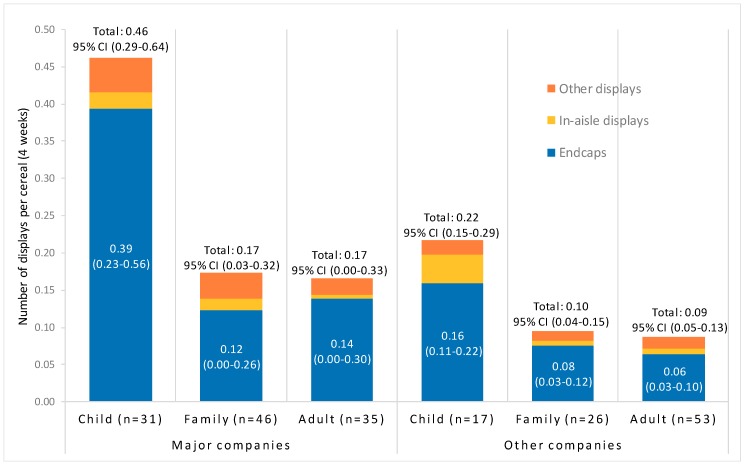
In-store displays per cereal over four weeks.

**Figure 3 ijerph-17-01284-f003:**
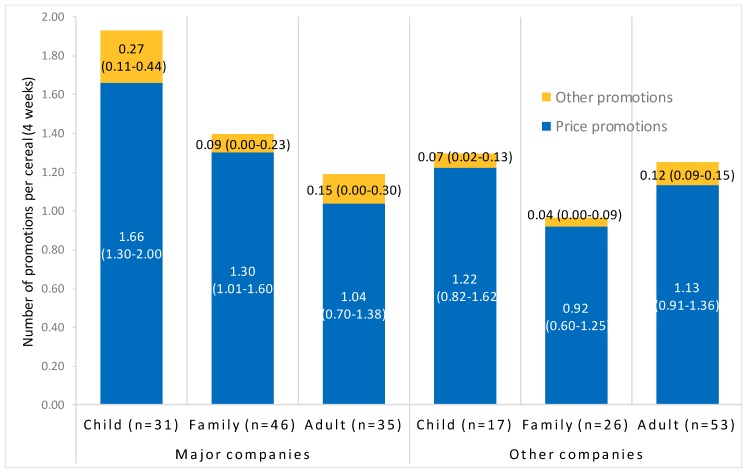
In-store promotions per cereal over four weeks.

**Figure 4 ijerph-17-01284-f004:**
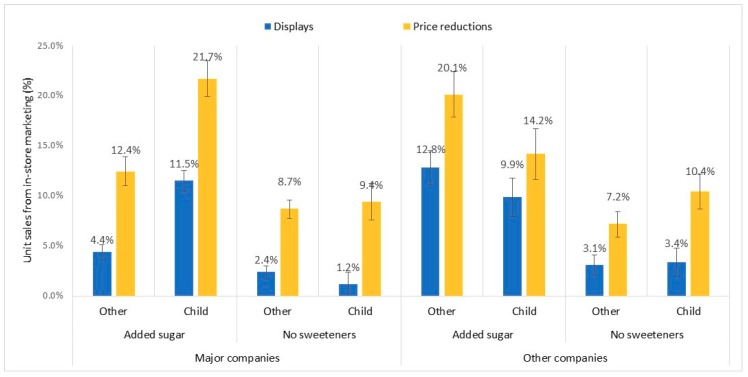
Percentage of sales due to displays and price promotions.

**Table 1 ijerph-17-01284-t001:** Child-targeted and other fruit drinks and juices.

Type of Company and Sweetener	Target Audience	
	Child-Targeted *N* = 733 (31.6%)	Other *N* = 1588 (68.4%)
Company type		
Major	428 (58.4%)	1048 (66.0%)
Other	305 (41.6%)	540 (34.0%)
Sweetener type		
Added sugar	509 (69.4%)	735 (46.3%)
No added sweeteners	224 (30.6%)	668 (42.1%)
Non-nutritive sweeteners only	0 (0.0%)	185 (11.6%)

Products with non-nutritive sweeteners were removed from the analyses (final *N* = 2136).

**Table 2 ijerph-17-01284-t002:** Legally viable policy options at the U.S. state level to regulate unhealthy food marketing in retail locations *.

Type of Policy	Examples
Information provision	Public service announcements
Product location in stores	Require that nutrient-poor products be stocked in less accessible locations, such as separate store aisles or specifically identified check-out aisles, and thus removed from prominent locations such as endcaps, freestanding and other special displays, store entrances, and general check-out aisles
Purchase restrictions	Institute age limits on purchases of harmful products (e.g., energy drinks)
Conditional licensing	Require retailers to meet conditions (e.g., product location, age limits) to maintain their license Utilize license fees to cover inspections to ensure compliance
Pricing	Excise tax on sales of certain products (e.g., sugary drinks) Set minimum price mandates and/or prohibitions on coupons and discounting (independently or in conjunction with a tax) Earmark tax revenues for health promotion, such as subsidizing fruits and vegetables

* These policies would apply to foods and beverages that do not meet minimum nutrition standards and/or products that are harmful to children’s health (e.g., sugary drinks, energy drinks).

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
