# Peer review of "Marketing to Children in Supermarkets: An Opportunity for Public Policy to Improve Children’s Diets"

_ijerph, 2020, doi:10.3390/ijerph17041284_

Round 1
Reviewer 1 Report
It was a great pleasure that I reviewed the manuscript entitled “Marketing to Children in the Supermarket: An Opportunity for Public Policy to Improve Children’s Diets.” I think the paper is very well-written and it presents very interesting findings. Here are several comments and suggestions that might help strengthen the paper. I am presenting those comments and suggestions in (mostly) chronological order.
In Abstract, to signify the importance of the studies reported, I would suggest to change Line 15 such that “in-store child-targeted marketing of nutrient-poor products in two U.S.-based studies.” The introduction section is very well-written. Only thing I would suggest is to clarify how the authors examine “impact of marketing for child-targeted products in the supermarket. It is basically described in the following section “1.3 The present research” but I think more details of the research paradigm should be helpful to readers. In the Materials and Methods section (2.1) for Study 1, the authors should provide more details of how 87 stores were chosen. Are they chosen randomly from 400 stores in the first week audits? Relatedly, the results reported in 2.2.1 are similar if the shelf facing data was analyzed among 87 stores. Also, Lines 195-200 should more clearly state that those percent values serve for DVs. Similarly, in the Materials and Methods section (3.1) for Study 2, the authors should provide more details for each DV – incremental sales due to displays and price reductions were operationally determined. More generally, it was surprising to know for some readers (at least myself) that child-targeted cereals and drinks are actually less healthier than those for adults. The authors provide two key references (52) and (57), and this study itself supported this, but is there any evidence? Relatedly, I would appreciate if the authors discuss possible reasons why this is the case in the Discussion section. By taking advantage of the loopholes, companies want to see child-targeted products by making less healthy while more tasty?
Author Response
It was a great pleasure that I reviewed the manuscript entitled “Marketing to Children in the Supermarket: An Opportunity for Public Policy to Improve Children’s Diets.” I think the paper is very well-written and it presents very interesting findings. Here are several comments and suggestions that might help strengthen the paper. I am presenting those comments and suggestions in (mostly) chronological order.
Thank you. We appreciate the positive feedback.
In Abstract, to signify the importance of the studies reported, I would suggest to change Line 15 such that “in-store child-targeted marketing of nutrient-poor products in two U.S.-based studies.”
We made the suggested change (Lines 15-16)
The introduction section is very well-written. Only thing I would suggest is to clarify how the authors examine “impact of marketing for child-targeted products in the supermarket. It is basically described in the following section “1.3 The present research” but I think more details of the research paradigm should be helpful to readers.
Thank you for this suggestion. We have added a sentence to the end of the introduction briefly describing the two types of data that we used for these studies (Lines 150-153).
In the Materials and Methods section (2.1) for Study 1, the authors should provide more details of how 87 stores were chosen. Are they chosen randomly from 400 stores in the first week audits?
We have added a sentence to clarify that these 87 stores were randomly selected from the original 400 stores to include one store from each supermarket chain in each of the 16 cities (see Lines 183-184).
Relatedly, the results reported in 2.2.1 are similar if the shelf facing data was analyzed among 87 stores.
Given the significant differences found in the larger sample of stores, we would expect to find these same differences in a smaller representative sample of these same stores. Since in-store promotions change week-to-week, we determined that it would be best to use our resources to obtain four weeks of data for a smaller sample of stores rather than obtain promotions data for the full sample of stores, but for a shorter period of time.
Also, Lines 195-200 should more clearly state that those percent values serve for DVs.
We now specifically state that the variables described in this paragraph are the DVs we used for analysis (lines 195 and 198).
Similarly, in the Materials and Methods section (3.1) for Study 2, the authors should provide more details for each DV – incremental sales due to displays and price reductions were operationally determined.
We have provided more detail about how IRi identifies sales due to displays and price reductions. We now also explain that these measures have been established by manufacturers and retailers to monitor their competitors’ in-store marketing (Lines 379-387, 393-394).
More generally, it was surprising to know for some readers (at least myself) that child-targeted cereals and drinks are actually less healthier than those for adults. The authors provide two key references (52) and (57), and this study itself supported this, but is there any evidence?
Yes, this finding is shocking to many people. In addition to references 52 and 57 that demonstrate the poor nutritional quality of children’s cereals and drinks, many studies have shown that virtually all products marketed to children are high in sugar, fat, sodium and/or calories, and this is why the WHO recommends that countries regulate food marketing to children. We have added two sentences to explain this point and reference a systematic review and the WHO recommendations in support (Lines 537-542).
Relatedly, I would appreciate if the authors discuss possible reasons why this is the case in the Discussion section. By taking advantage of the loopholes, companies want to see child-targeted products by making less healthy while more tasty?
To address this point, we have added a sentence that describes how industry self-regulation and repeated promised to market healthier foods to children have resulted in minimal improvements (Lines 542-544). We can only speculate on the reasons that companies choose to continue to market their least nutritious products directly to children as they do not openly discuss their marketing strategies, and their documents and own research findings are proprietary so not available to academic researchers. This question is fascinating, but would involve a lengthy discussion of profit motives and corporate ethics, which is beyond the scope of this paper.
Reviewer 2 Report
The paper contains interesting information about child-targeted marketing in two different studies. The research covers the comparison in-store marketing of children’s breakfast cereals with marketing of other (family/adult) and the proportion of product sales associated with in-store displays and promotions for child-targeted versus other fruit drinks/juices.
The structure of the paper do not meet the instructions for the authors given on the website – the editorial board makes last decision about it.
The confusion around the aim left me unclear whether this was intended to be a methodological paper. Please define the aim and justify why it is so significant on the current research.
The introduction section is to long. Some descriptions are not necessary.
The authors stated that are the first summarizing exinsting litarature on child-targeted marketing. The examples of papers are listed below.
- Hayeon Song, Bonnie Halvorsen, Amy Harley. Marketing cereal to children: content analysis of messages on children's and adults' cereal packages. International Journal of Consumer Studies, 2014, 38 6 571 – 577. https://doi.org/10.1111/ijcs.12116
- H. Brinsden, T. Lobstein . Comparison of nutrient profiling schemes for restricting the marketing of food and drink to children Pediatric Obesity, 2013, 8 4 325 – 337 https://doi.org/10.1111/j.2047-6310.2013.00167.x
Moreover, unfortunately, a lot of cited references are dated. I think, the manuscript should be enriched with present references. More them 34% references given in the paper are older than 10 years.
The weakest part of the paper is the discussion. Most described results are not discussed and only a small part of the paper can be accepted as a true discussion. For instance: store propotions per cereal over four weeks, the interaction between company type and target audience. And many of them. The sections 4.1 and 4.2 are not the discussion. The results need much more extensive discussion to be valuable to the reader and to gain true scientific value. Morever, it is not recommenended to use „as predicted” in the discussion section in the scientific paper.
An important case concerns the year of the survey. What is the reason to publish the results from 2009 (L. 179)?
The conclusions should be based on the results. At the moment a lot of conclusions are the statements. The conclusions do not tie together the elements of the paper. Please try to present it in a broader context. The conclusion section should be rewritten.
A regional study like this could provide a useful additional case study to extend the literature on child-targeted marketing; however, there are a number of weaknesses in the current article and it is not suitable for publication in its current form.
Author Response
The paper contains interesting information about child-targeted marketing in two different studies. The research covers the comparison in-store marketing of children’s breakfast cereals with marketing of other (family/adult) and the proportion of product sales associated with in-store displays and promotions for child-targeted versus other fruit drinks/juices.
Thank you.The structure of the paper do not meet the instructions for the authors given on the website – the editorial board makes last decision about it.
It is our understanding that there is flexibility in the structure of the paper, depending on the content of the research. We believe that this structure is appropriate for the topic of the special issue, “Marketing of Foods and Beverages: Impact and Potential Solutions for Children and Young People’s Health.”The confusion around the aim left me unclear whether this was intended to be a methodological paper. Please define the aim and justify why it is so significant on the current research.
We believe that this comment refers to the abstract, as we cite our specific hypotheses for the two research studies at the end of the introduction (section 1.3), where we also explain its significance. To address this concern, we revised the beginning of the abstract to cite the public health significance of unhealthy food marketing to children and how most current policies do not address in-store marketing (Lines 12-13). We also removed mention of the literature review in the abstract and describe the overall aims for the two research studies reported (Lines 15-16). In addition, we added a more general description of our primary aims at the end of the first part of the introduction (Lines 63-66) and expanded the sentence on the importance of this research (Line 146).The introduction section is to long. Some descriptions are not necessary.
As suggested, we removed some unnecessary detail from the first few paragraphs of the introduction (lines 38-57). We were unsure what unnecessary descriptions the reviewer is referring to in Sections 1.1 or 1.2. Currently, we have one short paragraph describing each relevant topic of the literature review on in-store marketing. We believe that this brief review of all relevant topic areas will be of value to the field as no previous research studies have consolidated these findings. To clarify why we believe this literature review is necessary, we now explain that this review addresses a gap in the existing research (Lines 59-60 and 114-115).The authors stated that are the first summarizing exinsting litarature on child-targeted marketing. The examples of papers are listed below.
- Hayeon Song, Bonnie Halvorsen, Amy Harley. Marketing cereal to children: content analysis of messages on children's and adults' cereal packages. International Journal of Consumer Studies, 2014, 38 6 571 – 577. https://doi.org/10.1111/ijcs.12116
- H. Brinsden, T. Lobstein . Comparison of nutrient profiling schemes for restricting the marketing of food and drink to children Pediatric Obesity, 2013, 8 4 325 – 337 https://doi.org/10.1111/j.2047-6310.2013.00167.x
We are not certain what statement the reviewer is referencing. Is this regarding this sentence on line 59-60 which read, “In this paper, we first summarize existing literature on child-targeted marketing in the supermarket”? We have reworded this sentence to clarify that we are discussing the order of the paper, not that we are the first researchers to summarize the existing literature on child-targeted marketing. In fact, throughout the paper we cite many prominent references that have previously summarized the literature on child-targeted marketing overall. We would like to again note that one reason we provide a thorough review of the literature on in-store marketing to children is that to our knowledge, no previous papers have reviewed this literature. Even extensive reviews, such as WHO (2010) and Cairns et al. (2013) do not discuss shelf placement, in-store displays, or price promotions.Moreover, unfortunately, a lot of cited references are dated. I think, the manuscript should be enriched with present references. More them 34% references given in the paper are older than 10 years.
We understand the reviewer’s point. We believe that this comment refers primarily to the references in Sections 1.1 and 1.2. We chose to cite these older studies for several reasons: 1) As noted earlier, there is relatively little existing research on child-targeted marketing at retailers in both the public health and marketing literature (Section 1.1). With the exception of child-targeted messages on product packages, few studies have examined this form of marketing, so we cite all available studies, even older ones, in this section. 2) There is also relatively little research on the effects of in-store marketing (Section 1.2). We have stated this point and include a 2014 review of the “shopper marketing” literature that also highlights this point (reference #40). 3) The studies on pester power represent the formative research on this topic. As there are relatively few studies on the topic, we believe that it is preferable to cite the primary research rather than secondary citations.The weakest part of the paper is the discussion. Most described results are not discussed and only a small part of the paper can be accepted as a true discussion. For instance: store propotions per cereal over four weeks, the interaction between company type and target audience. And many of them. The sections 4.1 and 4.2 are not the discussion. The results need much more extensive discussion to be valuable to the reader and to gain true scientific value. Morever, it is not recommenended to use „as predicted” in the discussion section in the scientific paper.
As is customary with scientific papers that report two separate studies, we include a discussion section following each results section (2.3 and 3.3). We believe that these sections explain the scientific significance for the specific findings in each of these studies, including the proportion of stores stocking different types of cereals (Lines 340-351) and the interactions between company type and target audience (e.g., Lines 360-363 and 510-518). Also as customary, the General Discussion (Section 4) synthesizes these findings and provides a broader overview of their significance. If the reviewer provides feedback on specific significant findings that were omitted from these discussion sections, we would be happy to include them. As requested, we removed “as predicted” from the first line of both discussion sections. We appreciate the reviewer’s point that sections 4.1 and 4.2 should not be included as part of the discussion. To address this concern, we have added a major heading 5 (Policy Implications) that includes the sections on self-regulation and government policies (now 5.1 and 5.2). Given the topic of this special issue, “Marketing of Foods and Beverages: Impact and Potential Solutions for Children and Young People’s Health,” we believe that this discussion of policy options to address in-store marketing to children is relevant and would be of interest to readers.An important case concerns the year of the survey. What is the reason to publish the results from 2009 (L. 179)?
We understand that the age of the data used in both studies is a limitation of this research. These two studies utilized existing available data to conduct extensive new analyses of data that had been reported previously only as topline findings. Since no previous research has examined the prevalence or potential impact of in-store displays or price promotions for children’s foods and only two studies have examined shelf placement, we believed that these analyses would provide important contributions to the literature. However, due to the considerable expense of conducting in-store audits and purchasing syndicated sales data, we would not have been able to commission another audit or purchase more recent data to conduct these analyses. We had previously indicated the age of the data to be a limitation of this research, but we have expanded on this limitation in the general discussion (Lines 547-550)The conclusions should be based on the results. At the moment a lot of conclusions are the statements. The conclusions do not tie together the elements of the paper. Please try to present it in a broader context. The conclusion section should be rewritten.
We appreciate this comment and agree that the last two paragraphs of the conclusion would be more appropriate to include in the previous section on policy options. We have moved the paragraph about industry self-regulation to the end of section 5.1 (Lines 583-593) and an abbreviated version of the paragraph about government regulation to the end of section 5.2 (663-667). We have kept the first paragraph of the Conclusion, as these statements are based directly on the study results. We also added a final sentence that refers to the need for government regulation to provide the broader implications of these results, as is customary for Conclusions (Lines 679-682).A regional study like this could provide a useful additional case study to extend the literature on child-targeted marketing; however, there are a number of weaknesses in the current article and it is not suitable for publication in its current form.
We hope that we have addressed the reviewers’ concerns and that it is now suitable for publication.Round 2
Reviewer 2 Report
The authors addressed my concerns and enhanced the paper. The necessary improvements has been done. But the conclusion section still need some small correction. Please delete the last sentence from the conclusion section. This is the conclusion written by Harris et al (2009). The conclusion should arise from your research (not other authors). I belive that you want to express the sentence: more effort should be done to regulate the common in-store marketing techniques for child-targeted products to avoid the epidemic of childhood obesity.
For future author's experience - please do not wait so long to publish the results. It is my first time when I accept the paper presenting so dated results. The reason of my decision are the results referencing to self placement, in-store displays and price promotions. If you published the results earlier, the government regulation concerning the child targeted marketing in supermarkets promoting the healthful diets for children would be active.
Author Response
We're pleased that you are satisfied with our revision. We have removed the final sentence as requested, and revised the previous sentence to add the following: "government regulation could be enacted to improve the supermarket environment to promote healthful diets for children." This revision is supported by our findings in section 5.2.